

# Spatial and temporal trends in western polecat road mortality in Wales

Allison Barg[1,2], Jenny MacPherson[3] and Anthony Caravaggi[1]

[1] Biological and Forensic Sciences, University of South Wales, Pontypridd, Wales, United Kingdom
[2] School of Natural Resources, University of Nebraska—Lincoln, Lincoln, NE, United States of America
[3] Vincent Wildlife Trust, Ledbury, Herefordshire, United Kingdom

## ABSTRACT

Roads have considerable ecological effects that threaten the survival of some species, including many terrestrial carnivores. The western polecat is a small-medium sized mustelid native to Asia and Europe, including Britain where its historical stronghold is in Wales. Polecats are frequently killed on roads and road casualties represent the most common source of data on the species in the UK. However, little is known about the factors that increase the risk of collision. We used Generalized Additive Models to explore seasonal patterns in collisions as well as using Principal Component Analysis and regression modelling to identify landscape characteristics associated with polecat road casualties in Wales. Polecat road casualties had a bimodal distribution, occurring most frequently in March and October. Casualties were more frequently associated with road density, traffic volume, presence of rabbits, habitat patchiness and the abundance of proximal improved grassland habitat. Casualties were negatively associated with elevation and the abundance of semi-natural grassland habitat. The results of this study provide a framework for understanding and mitigating the impacts of roads on polecats in their historic stronghold, hence has considerable value to polecat conservation as well as broader applicability to ecologically similar species.

## INTRODUCTION

There are currently almost 500,000 kilometres of road in the United Kingdom, over 3,000 kilometres of which has been built in the last 10 years (*UK Department for Transport, 2020*). As road networks continue to expand, it is critical to understand the impact that they have on wildlife populations (*Schwartz, Shilling & Perkins, 2020*). Roadways pose a serious risk to wildlife and can have a substantial impact on populations (*UK Department for Transport, 2019*; *Hill, De Vault & Belant, 2019*; *Schwartz, Shilling & Perkins, 2020*). For example, barrier effects caused by linear features in the environment—such as roads—can restrict animal movements (*Forman & Alexander, 1998*; *Schwartz, Shilling & Perkins, 2020*), potentially fragmenting populations into smaller, isolated subpopulations that may be more vulnerable to stochastic events (*Trombulak & Frissel, 2000*; *Jaeger & Fahrig, 2004*; *Riley et al., 2006*; *Klar, Herrmann & Kramer, 2009*).

Corresponding author
Allison Barg, allijoy15@gmail.com

Roads also have more direct effects in the form of injury and mortalities associated with wildlife-vehicle collisions (WVC). WVCs have been subject to increasing scrutiny in the last 20 years (*e.g.*, *Gomes et al., 2009*; *Langen, Ogden & Schwarting, 2009*; *Patrick, Gibbs & Popescu, 2012*; *Schwartz, Shilling & Perkins, 2020*; *Wright et al., 2020*) and are acknowledged as one of the most important sources of anthropogenic mortality of terrestrial vertebrates worldwide (*Hill, De Vault & Belant, 2019*). The frequency of WVCs is affected by road characteristics such as road class (*Clevenger, Chruszcz & Gunson, 2002*), traffic volume (*Jaarsma, Van Langevelde & Botma, 2006*) and speed limit (*Barrientos & Bolonio, 2009*), as well as surrounding land cover (*Wright et al., 2020*), roadside topography (*Clevenger, Chruszcz & Gunson, 2002*), species ecology and behaviour (*Grilo, Bissonette & Santos-Reis, 2009*; *Jacobson et al., 2016*) and resource abundance (*e.g.*, prey in roadside verges, *Planillo & Malo, 2013*). WVC 'hotspots' (*i.e.*, areas with high numbers of collisions) are often identified and used to target mitigation measures aimed at reducing the frequency and severity of collisions (*Langen, Ogden & Schwarting, 2009*; *Patrick, Gibbs & Popescu, 2012*; *Schwartz, Shilling & Perkins, 2020*; *Wright et al., 2020*), though the efficacy of this approach may be dependent on road age, (*Zimmermann Teixeira et al., 2017*). Road mortality data can also further our understanding of species distributions and movements (*Schwartz, Shilling & Perkins, 2020*). Threatened mammal taxa include felids (*Parchizadeh et al., 2018*; *Blackburn et al., 2021*), ursids (*Ha, 2021*), ungulates (*Putman, 1997*; *Riginos et al., 2022*), and mustelids (*Clarke, White & Harris, 1998*; *Russo et al., 2020*). Carnivore populations may be particularly susceptible to the negative effects of roads due to their unique life-history traits. For example, predators tend to be highly mobile, occur at low densities and have low reproductive rates (*Grilo, Smith & Klar, 2015*; *Ceia-Hasse et al., 2017*). Hence, they are and more likely to interact with a road network and a single mortality will have a greater impact on the population than would be the case for a more fecund species.

Western polecats (*Mustela putorius*) are small-medium (600 –1,500 g) mustelids native to Europe and Asia. They are considered generalists in both diet and habitat, although in the UK and Mediterranean regions they rely heavily on European rabbits (*Oryctolagus cuniculus*) and their burrows for both food and shelter (*Birks & Kitchener, 1999*; *Barrientos & Bolonio, 2009*). Polecats were once abundant in Britain but were hunted to extinction in Scotland and much of England in the 19th century, leaving remnant populations in some English counties and a stronghold in mid-Wales (*Langley & Yalden, 1977*; *Blandford, 1987*; *Birks & Kitchener, 1999*; *Packer & Birks, 1999*; *Birks, 2008*; *Croose, 2016*). Aided by the start of the first World War and the associated decrease in hunting and gamekeeping pressure, along with the recovery of the rabbit population post-myxomatosis (*Costa et al., 2013*), the Welsh polecat population rebounded (*Langley & Yalden, 1977*). Polecats have been listed on schedule 6 of the Wildlife and Countryside Act, which protects listed species from being killed or taken by certain methods, such as self-locking snares, since 1981 (*Great Britain, 1981*; Wildlife and Countryside Act 2020). They are also a priority species on the UK Biodiversity Action Plan (BAP) since 2007 (*BRIG, 2007*). Polecats have now successfully repopulated every county in Wales and much of southern England (*Croose, 2016*). There is the threat, though, of hybridisation with feral ferrets in England which may mask the true

distribution, making the Welsh population critical to maintaining the genetic legacy of the species (*Costa et al., 2013*).

Polecats exhibit several of the life-history traits that make many carnivores vulnerable to the impacts of roads (*Grilo, Bissonette & Santos-Reis, 2008*; *Ceia-Hasse et al., 2017*). They are also known to commonly consume carrion, which may put them at particularly high risk of WVC as roads and roadsides provide ample opportunities for scavenging (*Forman & Alexander, 1998*; *Grilo, Bissonette & Santos-Reis, 2009*; *Barrientos & De Dios Miranda, 2012*). Conversely, *Grilo, Bissonette & Santos-Reis (2008)* and *Grilo, Bissonette & Santos-Reis (2009)* suggest that polecats may be less susceptible to WVC than other carnivores due to an observed tendency to actively avoid roads. This type of discrepancy demonstrates the need for further, focused research on the topic.

Little is known about landscape characteristics that might increase the likelihood of WVCs involving polecats. The few studies that have been conducted thus far suggest that relevant factors include the speed and density of traffic (*Barrientos & De Dios Miranda, 2012*), proximity to rabbit burrows (*Barrientos & Bolonio, 2009*), presence of arable land and human settlements, and proximity to water courses or other linear features (*Červinka et al., 2015*). Here, we aimed to describe the temporal and spatial distribution of polecat WVC in Wales and to identify landscape characteristics associated with increased risk.

## MATERIALS AND METHODS

### Study area

Wales is a small country (20,782 km$^2$) of varied geography, with coastal plains in the south and west giving way to a largely mountainous interior dissected by deep rivers and valleys. Elevation ranges from 200 m above sea level on the coast to 1,085 m at the tallest peak. The landscape is primarily a matrix of natural and improved grasslands and pastures, with the exception of the mountainous areas in the centre and north of the country that are characterised by a mosaic of woodland, heathland and wetlands. It has a maritime climate; rainfall is common throughout the year, with an average of 1,000 millimetres annually (*Mayes, 2013*). The average daytime temperature ranges from 4 °C in the winter to 16 °C in the summer (https://www.metoffice.gov.uk/), though there is substantial spatial variation.

### WVC data

Observations of polecat road mortalities in Wales were provided by the Vincent Wildlife Trust (VWT). Records were collected *via* a national monitoring survey carried out from December 2013 to December 2015 (*Croose, 2016*). The survey used a community science framework to obtain reports of polecat casualties from members of the public, volunteers, and VWT members. Where possible, reports were verified by professionals with sufficient experience and expertise as polecats, feral ferrets or ferret-polecat hybrids *via* photo, video, or evaluation of the carcass (*Croose, 2016*). Both phenotypic and genetic studies suggest that most polecats in Wales are true polecats, while hybridisation is more common in England (*Costa et al., 2013*; *Croose, 2016*). During the survey period there were 85 observations in Wales verified as true polecats, no confirmed records of hybrids or feral ferrets, and 86 unverified records. We did not have access to carcass materials, hence further verification

was not possible. Therefore, unverified records were retained in analysis and assumed to be true polecats, giving a total of 171 observations. Due to the lack of true absence data a total of 1,200 pseudo-absence points (*Barbet-Massin et al., 2012*) were generated across the Welsh road network and at least 3 kilometres (*i.e.,* the average home range size of polecats in Britain; *Birks & Kitchener, 1999*) to close the parenthesis starting (*i.e.,* distant from the nearest presence point (Fig. 1).

## Environmental parameters

Species abundance is an important driver of road-kill patterns (*Barrientos & Miranda, 2012*; *Husby, 2016*). While the range and overall population-level abundance and distribution of polecats in the UK has increased, suitable abundance or density data are unavailable for polecats in Wales (*Mathews et al., 2018*). Simple population-occurrence relationships, where changes in habitat quality and/or quantity impacted the populations of associated species (*Freckleton, Noble & Webb, 2006*) offer a potential proxy for abundance. However, such models have never been applied to polecats in the UK and the extent to which they may be (un)suitable is unknown. Hence, our models did not capture local abundance or a proxy thereof.

Explanatory environmental variables were selected *a-priori*, based on previously published literature and knowledge of the species' ecology. These included proportional land cover, habitat patchiness, elevation, waterway density, road density, presence of rabbits and traffic volume (Table 1). The percentage of each land class and the total number of patches (where more patches = more fragmentation) were extracted from the 25 m resolution Land Cover Map 2015 (*UK Centre for Ecology and Hydrology, 2017*) within a 1.5 kilometre buffer of each presence and pseudo-absence point, based on average polecat home range size (*Birks & Kitchener, 1999*). In order to evaluate the effect of many land classes in fewer variables, classes of interest were combined using Principal Component Analysis (PCA) with Box–Cox transformation, prior to analysis. Three principal components (PCs) with eigenvalues >1 were retained for use in models. Elevation was extracted in R using the *elevatr* package (*Hollister et al., 2021*). We attempted to capture spatial autocorrelation using the *glmmPQL* function in the *MASS* package (*Venables & Ripley, 2002*), using a correlation structure (*Dormann et al., 2007*). However, all models returned zero-distance errors that resisted resolution. We also explored the use of GAMMs *via* the *mgcv* package (*Wood, 2003*; *Wood, 2011*; *Wood, 2017*), where spatial coordinates were included in a bivariate spline with a Markov random field smoother. These models performed very poorly. Hence, spatial autocorrelation in the data was captured by calculating Moran's I scores for each presence and pseudo-absence point using the Anselin's Local Moran's I tool within the Spatial Analyst toolbox in ArcGIS Desktop (Esri, Redlands, CA, USA; https://support.esri.com/en/Products/Desktop/arcgis-desktop/arcmap/10-8). All numeric variables were rescaled so that $=x = 0$, $\sigma = 1$ to facilitate direct comparison between covariates.

Prey abundance and availability are a key consideration when evaluating predator occurrence (*e.g.*, *Šálek et al., 2010*; *Mougeot et al., 2019*). Indeed, the proximity of rabbits to roads may be a reliable indicator of polecat collision risk (*Barrientos & Bolonio,*

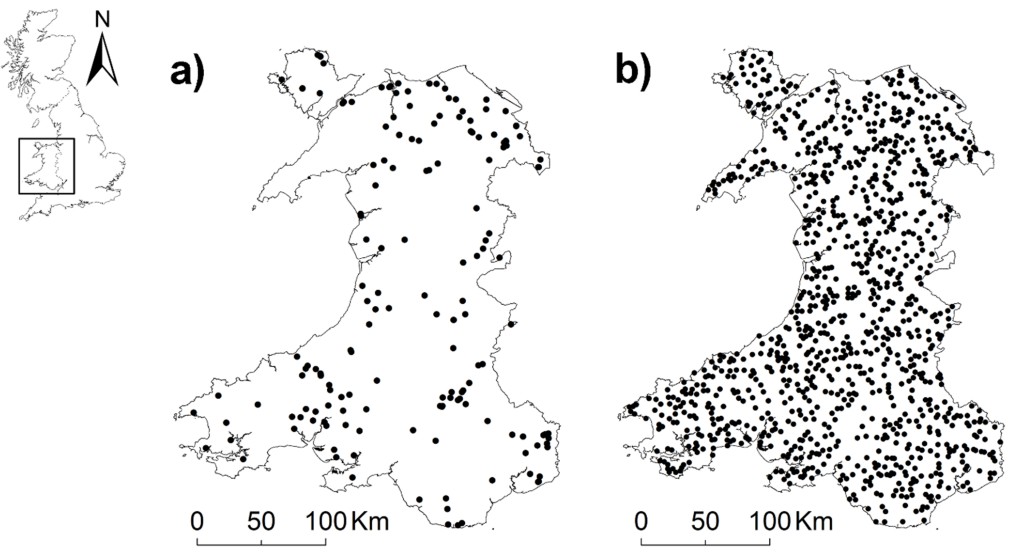

**Figure 1** Maps of Wales showing locations of (A) mortalities (*n* = 171) and (B) pseudo-absences (*n* = 1200). Country shapefiles were extracted using the *maps* (*Becker & Wilks, 2021*) and *mapdata* (*Becker & Wilks, 2018*) packages in R.

**Table 1** Summary of the environmental covariates used for modelling polecat road mortality. TC = Land Class Map 2015 target class.

| Description | Type | Source | Reference |
|---|---|---|---|
| Broadleaf woodland (% cover) | Raster, non-composite; TC 1 | Land Cover Map 2015 | *Baghli, Walzberg & Verhagen (2005)* |
| Coniferous woodland (% cover) | Raster, non-composite; TC 2 | Land Cover Map 2015 | *Zabala, Zuberogoitia & Martínez-Climent (2005)* |
| Arable (% cover) | Raster, non-composite; TC 3 | Land Cover Map 2015 | *Birks (2008)* |
| Improved grassland (% cover) | Raster, non-composite; TC 4 | Land Cover Map 2015 | *Baghli, Walzberg & Verhagen (2005)* |
| Semi-natural grassland (% cover) | Raster, composite; TC 5, 6, 7 | Land Cover Map 2015 | *Baghli, Walzberg & Verhagen (2005)* |
| Heathland (% cover) | Raster, composite; TC 9, 10 | Land Cover Map 2015 | *Fournier et al. (2007)* |
| Wetland (% cover) | Raster, composite; TC 8, 11 | Land Cover Map 2015 | *Birks (2008)* |
| Urban (% cover) | Raster, composite; TC 20, 21 | Land Cover Map 2015 | *Baghli, Walzberg & Verhagen (2005)* |
| Number of habitat patches | Raster, non-composite | Land Cover Map 2015 | *Zabala, Zuberogoitia & Martínez-Climent (2005)* |
| Density of roads | Spatial line | Open Street Map | *Forman & Alexander (1998)* |
| Density of water features | Spatial line | Open Street Map | *Baghli, Walzberg & Verhagen (2005)* |
| Distance to rabbits | Spatial point | National Biodiversity Network | *Barrientos & Bolonio (2009)* |
| Road type | Spatial line | Open Street Map | *Barrientos & Miranda (2012)* |
| Traffic volume | Spatial point | Department for Transport | *Langevelde & Jaarsma (2004)* |
| Elevation | Spatial point | Calculated with Elevatr in R | *Birks (2012)* |

*2009*). However, accurate, landscape-scale data on the presence or absence of many UK mammals—including European rabbits—are sparse (*Mathews et al., 2018b*). We therefore captured the potential influence of rabbits on polecat WVC using National

Biodiversity Network (NBN; https://nbn.org.uk/) rabbit data between 2010–2019 inclusive and calculating the minimum distance between mortalities/pseudo-absences and rabbit locations. The selected date range ensured national coverage and assumes that a rabbit observation in a given year represents a population that was present during the period 2013–2015 (*i.e.,* when polecat data were collected).

## Modelling seasonal trends

Seasonal trends in polecat WVC were analysed using a Generalized Additive Model (GAM), following *Wright et al. (2020)*. Five candidate models were created with the number of roadkill observations as the response variable and month as the predictor. Models used a Poisson family error distribution with a log-link function, and a cyclic cubic spline with 12 knots applied to the month variable. Temporal non-independence in the data was accounted for by including a correlation argument in four of the models where the autoregressive order, $p$ varied between 1–4. The fifth model assumed independence between observations. Residuals of each model were evaluated using the Auto-Correlation Function (ACF) and Partial Auto-Correlation Function, and the best-approximating model was identified using Akaike's Information Criterion (AIC).

## Modelling environmental relationships

The relationship between polecat WVC and percent land cover classification ('habitat model', hereafter), was explored using a Generalized Linear Model (GLM) using a binomial error distribution with logit link function. Presence/pseudo-absence of polecats was the dependent variable and the proportional land cover, patchiness, elevation, road density, road class, distance to rabbits, and Moran's I were fixed explanatory variables. All variables had Variance Inflation Factors (VIF) $\leq 2$ and were retained (*Zuur, Ieno & Elphick, 2010*). All possible model permutations of the resulting GLM were generated using the *MuMIn* package (*Barton, 2020*) and Akaike weights were calculated for each model within the top subset of models ($\Delta$AIC $\leq 2$). Both the best-approximating model ($\Delta$AIC $= 0$) and the full average model (*i.e.,* averaged across the top subset of $n$ models) were subsequently identified. Using both approaches accounts for uncertainty in model selection and produces more robust results than using the best-approximating model alone (*Grueber et al., 2011*).

A second GLM was produced using the same method and based on the subset of the data that included traffic volume ('traffic models', from hereon). Traffic volume data for 2015 (point locations) were obtained from the Department for Transport (*UK Department for Transport, 2019*), but data were not available for every road and, by extension, every presence/pseudo-absence location. However, traffic volume has frequently been reported as an important factor in determining risk of WVC (*Forman & Alexander, 1998*; *Van Langevelde & Botma, 2006*; *Jacobson et al., 2016*). Therefore, this second set of models was constructed using a reduced dataset ($n = 150$ presence; 1,058 pseudo-absence) containing only presence/pseudo-absence points on roads that were also associated with traffic volume data. WVC data did not necessarily directly overlap roads due to imprecision in location information, hence presence data were aligned with the nearest road, according to a 1.5-kilometer threshold.

All analyses were conducted in R (v.3.6.3; *R Core Team, 2019*). Code are available at https://zenodo.org/record/7025191 (*Barg, MacPherson & Caravaggi, 2022*).

## RESULTS

Of the 171 road mortalities identified and included in these analyses, 85 were considered to be true polecats and 86 were unverified (*i.e.,* plausible but not confirmed). WVC were most frequently recorded on trunk roads (28%), followed by unclassified roads (18%), primary roads (15%), and secondary roads (15%). Only one carcass was recorded on a motorway (Fig. 2).

Out of five candidate seasonal models, the model that performed best was the one that did not include a correlation argument. This model showed significant temporal variation across months (Intercept $\beta = 1.90 \pm 0.16$; s(month) $F = 2.78$, $p = 0.006$). WVC had a bimodal distribution, peaking in March and October. Winter WVC was substantially lower than summer WVC (Fig. 3).

PC1 accounted for 24% of total variance in the data and was positively associated with semi-natural grassland, and negatively associated with improved grassland. The loadings of PC2 (14%) described a negative association with improved grassland and a positive association with urban cover. PC3 (13%) was associated positively with wetland and heathland, and negatively with broadleaf and coniferous forest (Table 2).

The best-approximating habitat model suggested that polecat WVC occurred with greater frequency in landscapes that were more fragmented (habitat patchiness; $\beta = 11.57 \pm 1.84$ [95% Confidence Interval]) and had greater road densities ($\beta = 0.4.20 \pm 1.63$) than areas where WVC were not recorded. Polecat WVC were also more likely where rabbits occur (distance to rabbits; $\beta = 3.63 \pm 1.31$), at lower elevations ($\beta = -2.09 \pm 1.03$) and in areas with more semi-natural grassland and less improved grassland (PC1; $\beta = -1.17 \pm 0.31$). Polecat WVC also exhibited spatial autocorrelation; carcasses were found nearer to other polecat carcasses than would have been expected by chance (Moran's I; $\beta = 4.92 \pm 4.19$). Six models were within $\leq 2$ $\Delta$AIC of the best-approximating model, hence were used to create the average model. Six variables occurred across all six models (weight = 1.00): road density; habitat patchiness; distance to rabbits; PC1; elevation; and Moran's I. PC2 occurred in three models (weight = 0.57), and waterway density and PC3 both occurred in two models (weight = 0.37 and 0.19, respectively; Table 3).

Traffic models behaved similarly to habitat models. Polecat WVC were again associated with a more fragmented landscape ($\beta = 1.14 \pm 0.19$); greater road densities ($\beta = 4.585 \pm 1.70$), proximity to rabbits ($\beta = 3.80 \pm 1.39$), lower elevations ($\beta = -2.42 \pm 1.13$), and semi-natural grassland and less improved grassland ($\beta = -1.12 \pm 0.32$). There was a positive association between traffic volume and WVC, with more carcasses being recorded on more active roads ($\beta = 2.77 \pm 1.99$). Similar results were observed in the average model, in which six variables—distance to rabbits, elevation, habitat patchiness, road density, traffic volume, and PC1—occurred in all 14 models (*i.e.,* importance = 1) across the top subset of models. Waterway density (weight = 0.43), Moran's I (weight = 0.38) and PC3 (weight = 0.33) each occurred in six models (Table 4). It should be noted that our

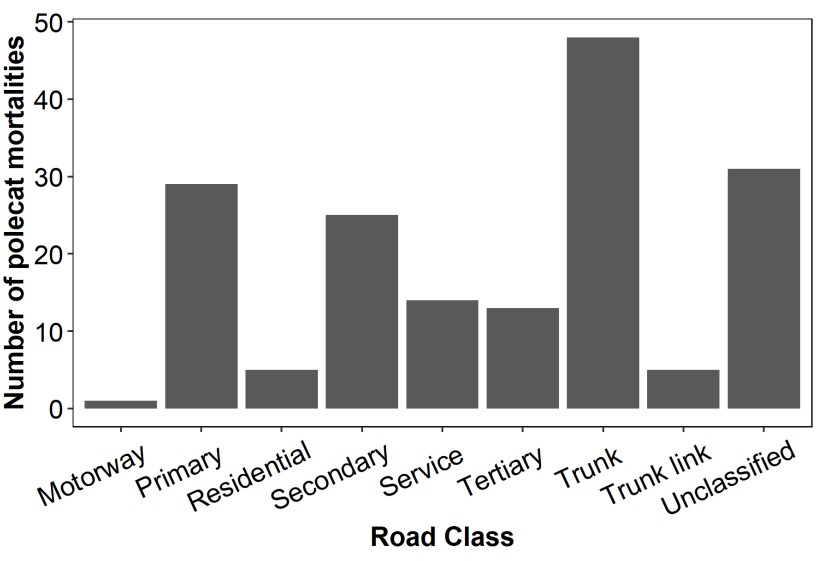

**Figure 2** Total numbers of polecat road casualties associated with certain road types in Wales.

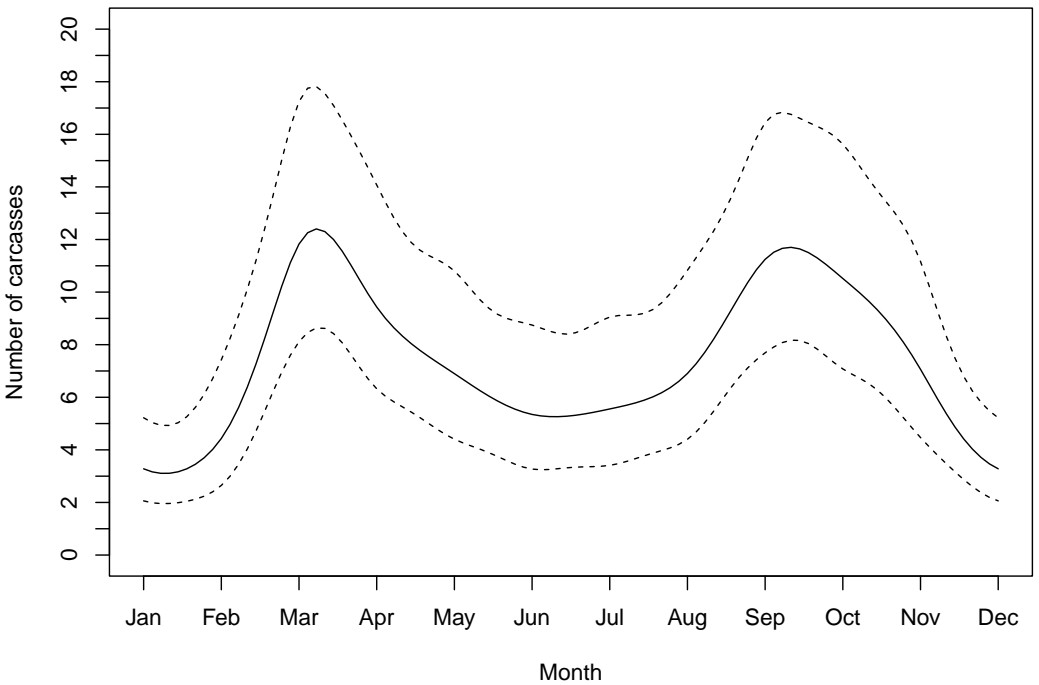

**Figure 3** Seasonal fluctuation in polecat WVC showing smoothed numer of detections with 95% confidence intervals.

**Table 2 Principal Component Axes loadings showing variation in the habitat classes used in roadkill models and retained within the top subset of models (Δ AIC ≤ 2; see Supplementary Information for PCs 4-8).** The percentage of total variation explained by each component is given in parentheses. Loadings that explain the largest proportion of each PC are in bold.

| Habitat type | Principal component axes | | |
|---|---|---|---|
| | PC1 (24%) | PC2 (14%) | PC3 (13%) |
| Arable | −0.297 | 0.012 | −0.360 |
| Broadleaf woodland | −0.277 | **0.392** | **0.427** |
| Coniferous woodland | 0.351 | 0.126 | **0.496** |
| Heathland | 0.263 | 0.070 | **-0.467** |
| Improved grassland | **−0.566** | **−0.411** | 0.054 |
| Semi-natural grassland | **0.540** | −0.186 | 0.051 |
| Urban | −0.115 | **0.787** | −0.173 |
| Wetland | 0.129 | 0.057 | **-0.435** |

**Table 3 Results of habitat models (n = 171 presence, 1200 pseudo-absence) investigating habitat and road characteristics in relation to polecat road mortality.** The conditional average model was created from the top subset (Δ AIC ≤ 2) of six models. Regression coefficients (β) and 95% confidence intervals (±95% CI) are given as well as the significance of each variable where * $P < 0.05$, ** $P < 0.01$, and *** $P < 0.001$. The weight of variables included in conditional average model is also given, with the importance (number of candidate models that contain the variable) in parentheses.

| Variable | Best-approximating model | | Full average model | | |
|---|---|---|---|---|---|
| | β | ±95% CI | β | 95% CI | Weight |
| Habitat patchiness | 11.57 | 1.84*** | 11.47 | 1.83*** | 1.00 (6) |
| Moran's I | 4.92 | 4.19* | 4.89 | 4.16* | 1.00 (6) |
| Road density | 4.20 | 1.63*** | 4.19 | 1.63*** | 1.00 (6) |
| Distance to rabbits | 3.63 | 1.31*** | 3.66 | 1.33*** | 1.00 (6) |
| Elevation | −2.09 | 1.03*** | −2.09 | 1.06*** | 1.00 (6) |
| PC1 | −1.17 | 0.31*** | −1.17 | 0.31*** | 1.00 (6) |
| PC2 | −0.18 | 0.22 | −0.10 | 0.23 | 0.57 (3) |
| Waterway density | – | – | 0.47 | 1.67 | 0.37 (2) |
| PC3 | – | – | −0.02 | 0.03 | 0.19 (2) |

Notes.
*$P < 0.05$.
**$P < 0.01$.
***$P < 0.001$.

thinned traffic dataset was biased towards trunk roads and primary roads, both meant for high volumes of traffic. Almost all records associated with these road classes were retained, compared to only 10% of records from unclassified roads, 15% from service roads and 30% from tertiary roads.

## DISCUSSION

Polecat WVC occurred most frequently in March and October, coinciding with the seasons that polecats are most mobile (*Birks, 2012*). Prior research suggests that carnivores

**Table 4** **Results of traffic models analyzing habitat and road characteristics related to polecat road casualties using only data points associated with traffic counts ($n = 150$ presence, 1,058 pseudo-absence).** The conditional average model was created from the top subset ($\Delta$ AIC $\leq 2$) of fourteen models. Regression coefficients ($\beta$) and 95% confidence intervals ($\pm$95% CI) are given as well as the significance of each variable where * $P < 0.05$, ** $P < 0.01$, and *** $P < 0.001$. The weight of variables included in conditional average model is also given, with the importance (number of candidate models that contain the variable) in parentheses.

| | Best- approximating model | | Full average model | | |
|---|---|---|---|---|---|
| Variable | $\beta$ | 95% CI | $\beta$ | 95% CI | Weight |
| Road density | 4.58 | 1.70*** | 4.05 | 0.83*** | 1.00 (14) |
| Distance to rabbits | 3.80 | 1.39*** | 3.86 | 1.39*** | 1.00 (14) |
| Traffic volume | 2.77 | 1.99** | 2.49 | 2.02* | 1.00 (14) |
| Elevation | −2.42 | 1.13*** | −2.41 | 1.13*** | 1.00 (14) |
| Patches | 1.14 | 0.19*** | 1.13 | 0.19*** | 1.00 (14) |
| PC1 | −1.12 | 0.32** | −1.13 | 0.32*** | 1.00 (14) |
| PC2 | −0.20 | 0.26 | −0.11 | 0.28 | 0.55 (7) |
| Waterway density | – | – | 0.56 | 0.90 | 0.43 (6) |
| Moran's I | – | – | 1.12 | 4.11 | 0.38 (6) |
| PC3 | – | – | −0.05 | 0.20 | 0.33 (6) |

**Notes.**
* $P < 0.05$.
** $P < 0.01$.
*** $P < 0.001$.

are particularly vulnerable during specific life-history stages (*Grilo, Bissonette & Santos-Reis, 2009*). Polecats typically occupy fixed home ranges throughout the year, with two exceptions—the breeding season (March and April) and during the period of kit dispersal in autumn (*Birks, 2012*). Polecats are polygynous (*Lodé, 2001*) and males travel outside of their home-range to find and mate with several females each year, likely increasing their interactions with roads. Moreover, juveniles leaving their natal home range may be particularly susceptible to WVC as they are likely naïve to the variable dangers of roads (*Grilo, Smith & Klar, 2015*; *Carvalho et al., 2018*). High juvenile WVC in the autumn has the potential to negatively impact recruitment and perturb population age- and genetic structure (*Holderegger & Di Giulio, 2010*). We recommend that future WVC studies aim to collect age and sex information whenever possible with the goal of exploring potential population-level impacts.

There was a strong relationship between polecat WVC and the presence of European rabbits adjacent to the road. Rabbits make up the majority of polecat diet in Wales, as well as other countries with large rabbit populations, and polecats frequently use rabbit warrens as daytime resting sites (*Birks & Kitchener, 1999*; *Barrientos & Bolonio, 2009*; *Birks, 2012*). Prey availability in roadside verges has been linked to increases in predator roadkill, especially in heavily agricultural or developed areas where verges may provide refuge for small and medium sized mammals (*Barrientos & Bolonio, 2009*; *Silva et al., 2019*). The presence of rabbits near roads may also lead to higher rabbit road mortality, offering scavenging opportunities to polecats and other species, which may prove dangerous (*Silva et al., 2019*).

Habitat analysis showed that WVC was positively associated with areas of improved grassland, and negatively with semi-natural grassland. Grassland is the most prominent land classification in Wales, and is the preferred habitat for European rabbits (*Birks, 2012*; *Mathews et al., 2018b*). The split between improved and semi-natural grassland is likely an effect of elevation, as semi-natural grassland is found primarily in upland areas and improved grassland in lowland areas. The negative relationship with elevation also appeared in the models. Several past distribution surveys have reported that polecats in the UK are less common in upland areas than lowlands, potentially due to a reduction in available resources at higher elevations (*Birks & Kitchener, 1999*; *Croose, 2016*). *Birks & Kitchener (1999)* made the connection that roadways tend to follow valley bottoms and suggested the increased road density in lowland areas as a potential reason that polecats have been slow to recolonise parts of the South Wales valleys, the most densely populated part of the country. The lack of density estimates for polecats, or a reliable proxy thereof, meant that population metrics were not included in our models. However, polecat densities are thought to be uniform across habitats in Wales, with the exception of urban areas (*Mathews, Wright & Kendall, 2018*; *Mathews et al., 2020*), so the trends observed here are unlikely to be due to differing densities across habitat types. Nevertheless, we recommend further surveys focused on determining polecat population metrics to inform future models.

The influence of habitat patchiness evident herein suggests that polecats are at greater risk in heavily fragmented landscapes. Several studies on polecat habitat use have shown a preference for heterogeneous landscapes with a high number of habitat patches (*Zabala, Zuberogoitia & Martínez-Climent, 2005*; *Mestre, Ferreira & Mira, 2007*). Indeed, some degree of fragmentation can be beneficial for a generalist predator such as the polecat as it may provide a higher variety of food resources and a large amount of edge habitat, which can increase landscape-scale connectivity (*Zabala, Zuberogoitia & Martínez-Climent, 2005*). However, fragmentation and interactions that increase WVC risk can create barriers to movement and impede functional connectivity (*Forman & Alexander, 1998*; *Grilo, Studies & Bissonette, 2011*). Further, the influence of road density on polecat WVC risk is clear. It has been suggested that polecats actively avoid roads (*Grilo, JA & Santos-Reis, 2008*; *Grilo, JA & Santos-Reis, 2009*). However, roadside vegetation can provide cover for polecats as well as ample opportunities for scavenging roadkill (*Schwartz et al., 2018*). Polecat population density may also play a role, as high densities of impacted wildlife species have been shown to be important predictors of WVCs in some studies (*e.g.*, *Rolley & Lehman, 1992*; *Mayer et al., 2021*). Thus, active avoidance is unlikely to be possible where roads or polecats occur at sufficient densities that interactions are inevitable.

We found that higher traffic densities were associated with increased frequencies of polecat WVC. This corresponds with previously published literature on WVC in polecats and other species (*Grilo, Bissonette & Santos-Reis, 2009*; *Červinka et al., 2015*; *Jacobson et al., 2016*). The relationship between WVC and traffic volume (traffic flow theory) suggests that collisions will increase in accordance with traffic volume to a certain threshold, after which high traffic activity causes animals to avoid roads entirely (*van Langevelde & Jaarsma, 2004*; *Moore et al., 2020*). *Jacobson et al. (2016)* expanded on the traffic flow model using species-specific behavioural responses to traffic to better predict the effect of WVC and

barrier effects on populations. Within their framework species are categorized into four categories: nonresponsive, pausers, speeders and avoiders. Polecats most likely fit into the pauser category along with skunks and porcupines. This group tends to respond to perceived risk using by reducing their speed or freezing, increasing the time spent on the roadway and increasing risk of WVC especially in high traffic areas (*Jacobson et al., 2016*).

WVC were most often reported on trunk roads and primary roads, both road classes meant for long-distance travel and associated with high traffic volume and high speed limits (*UK Department for Transport, 2012*). There is likely an element of observational frequency bias in these results, where a greater number of people using a road leads to a greater number of roadkill reports. However, the relationship between WVC and roads with high traffic volume and speeds is well documented (*van Langevelde & Jaarsma, 2004*; *Jaarsma, Van Langevelde & Botma, 2006*; *Pagany & Dorner, 2016*; *Canal et al., 2019*; *Russo et al., 2020*). Second to trunk roads, the greatest number of observations came from unclassified roads, which are small roads connecting rural and suburban areas that make up approximately 60% of the roads in the UK (*UK Department for Transport, 2012*). While this may appear to contradict the result that WVCs increase with traffic volume, the sheer number of unclassified roads likely contributed to the comparatively high number of recorded collisions. Reports may also be somewhat biased towards unclassified roads as, although WVC are less frequent, they may be more likely to be reported when they do occur. Unclassified roads generally have slow speed limits and little traffic, making it easier to spot roadkill while driving and more convenient to stop, examine, and report the carcass than it would be on a busy road.

## CONCLUSION

This paper is the first to describe factors influencing polecat WVC in Wales, the historic stronghold for the species in Britain. As polecat populations continue to recover after near extirpation, addressing potential risks to their survival is crucial. The Welsh polecat population is important to maintaining the genetic legacy of polecats in Britain, as it has been shown that they have higher rates of hybridisation with feral ferrets elsewhere in their range. Further research on this topic should look at population-level effects of WVC and spatial variation thereof and seek to identify appropriate mitigation measures. This research has implications for polecat protection in Wales as well as applicability to the rest of Britain, and throughout their European range, where they are thought to be in decline in several countries.

## ACKNOWLEDGEMENTS

The authors would like to thank the Vincent Wildlife Trust for providing data on polecat WVC, as well as the many volunteers who submitted observations to make this work possible.

### Funding

The authors received no funding for this work.

### Competing Interests

The authors declare there are no competing interests.

### Author Contributions

- Allison Barg conceived and designed the experiments, performed the experiments, analyzed the data, prepared figures and/or tables, authored or reviewed drafts of the article, and approved the final draft.
- Jenny MacPherson conceived and designed the experiments, performed the experiments, authored or reviewed drafts of the article, and approved the final draft.
- Anthony Caravaggi conceived and designed the experiments, performed the experiments, analyzed the data, prepared figures and/or tables, authored or reviewed drafts of the article, and approved the final draft.

### Data Availability

The data and code are available at Zenodo: Allison Barg, Jenny MacPherson, & Anthony Caravaggi. (2022). arcaravaggi/Barg_polecats: Code archive - Spatial and temporal trends in western polecat road mortality in Wales. In PeerJ (v0.0.1). Zenodo. https://doi.org/10.5281/zenodo.7025191.

### Supplemental Information

Supplemental information for this article can be found online at http://dx.doi.org/10.7717/peerj.14291#supplemental-information.

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
