# Peer review of "Spatial and temporal trends in western polecat road mortality in Wales"

_PeerJ, doi:10.7717/peerj.14291_

## Round 0.1 · original submission · Major Revisions

Please follow the detailed comments made by the Reviewers, especially in the analysis and structure of the paper. I hope you find it suitable to do I am sure it will improve your paper

Reviewer 1 ·

Basic reporting

Some section of the manuscript were ambiguous and the terminology used confused the point. I have made extensive suggestions on way in which this could be improved.

The literature reviewed is largely relevant, however, very specific to the study site and there is a lack of broader literature that would better contextualise this work in the international literature. Much of the language assumes a knowledge of the UK, and Wales legislation etc. This is not ideal for international readers such as myself.

Presentation of Table 3 could be improved, unclear the purpose of the numbers in brackets next to AIC weights. Inclusion of significance levels against SE is not common; I would recommend calculating 95% confidence intervals and then emboldening those that are important.

Figure 1 - need to include the road network on this figure.
Figure 2 - this looks overly convoluted (possibly overfit a little because of the 12 knots) This figure needs improvement before publication.
Figure 3 - this should be addressed and described in the results section.

Experimental design

The research questions and particularly justification of the variables and statistical approaches are needed.

The analytical methods are appropriate for the data, though the way in which they're report suggests they may not be the most confident with analysis and there is work needed to most appropraite report the statistics used.

The results section needs reworking as the authors confuse two statistical philosophies and don't adequately explain their results with trends in the data often left undescribed. This is particularly noticable for the PC variable which warrants considerable explanation for the reader.

Validity of the findings

The results are interesting and will add to our knowledge of the impact of roads on polecats. More justification on the fundamental literature and more theoretical links in the discussion are important to more appropraitely contextualise the study.

Conclusions are a little vague and not overly supported by the data in all cases. The authors need to be more objective and critical of their data and what they show, particularly in the context of the nature of the input data which limits the inferential power.

Additional comments

Overall, this was an interesting manuscript to read. I suspect this is a good first paper of a PhD or graduated Honours student which is great to see, some work on re-focussing the paper to the core story and appropriate reporting of the results will help improve this manuscript ready for publication.

I have provided extensvie comments in the attached which I hope are helpful .

Annotated reviews are not available for download in order to protect the identity of reviewers who chose to remain anonymous.

·

Basic reporting

General comments:
The manuscript “Spatial and temporal trends in western polecat road mortality in Wales” brings an overview of polecat vehicle collisions in Wales and discussed temporal and spatial patterns, showing some important variables related to spatial distribution. In the whole parts of the manuscript (abstract, objectives, results, and discussion), the authors did not follow the same idea sequence. Sometimes they started with temporal analysis, sometimes with spatial findings, and the discussion only focused on spatial findings and did not bring any temporal discussion. It is important to standardize the presentation sequence to make reading easier. Methods must be clarified some important questions to facilitate comprehension. Examples: 1 - how reliable are the data used as there are some unverified data? 2 – Why was PCA used? 3 – Why did they separate habitat and traffic models since they used averaging models? There are some assumptions behind these choices, but they are not explicit. Also, the term ‘mortality’ should be revised, and other related terms should be standardized (WVC, casualty, mortality, roadkill). I made point-by-point comments:

Experimental design

Methods:
Line 107: Revise “mortality”. mortality is a population parameter. As we do not know exactly the effect of each casualty in the population, I recommend using casualties or fatalities instead of mortality. Or use WVC since you used it in the introduction. It is important to standard the terms in the whole manuscript.
Lines 112-113: Is the identification reliable? As I am not from England or Wales, I am not sure, but I imagine that the general population can misidentification polecats and ferrets. Thus, some records could be wrong. I worked with some road staff with some training on wildlife identification and sometimes there were some huge mistakes. One example is a study that showed that “Non-experts do not reliably identify species that resemble other species”: https://doi.org/10.1016/j.biocon.2018.06.019. Was there any record that was identified as a polecat and when verified it was not? If not, I recommend including this information to become the data more reliable. The unverified records were not possible to verify because they did not have photos? This assumption that all records are reliable should be clarified.
Line 120: why 1,200 points?
Line 122: why 3 kilometers?
Line 122: Standardize the units: 3,000 meters or 1.5 kilometers at line 120.
Line 129: the same: standardize 1.5 km ou 1,500 meters.
Line 145: Is ‘NBN’ the National Biodiversity Network? Include the whole name.
Line 150-158: Does species abundance refer to prey abundance (rabbit) or polecats? I understood that refers to polecats. It seems that the authors justified why they did not use abundance in that part of the methods. I am not sure if it is necessary. Maybe they can remove that part or replace it in the discussion section.
Line 166: why 1,058?
Line 171: the same as line 129: standardize 1.5 km ou 1,500 meters.
Line 174: It is not clear what variables were used to model temporal trends. Did you only use months as the predictor? If it is only months, why did you not use circular analysis, it seems that fix to your data.
Line 176: As they used Generalized additive mixed models, it is not clear which variable computed the random effects.
Line 176: Which are the five models?
Line 186: It is not clear which environmental variables were used. I imagine that they were those described on the “Environmental parameters” topic (land cover, habitat patchiness, elevation, waterway density, road density, presence of rabbits, and traffic volume). However, the term “percent land cover classification” makes me confused. Which were variables considered?
Line 203: I did not understand why authors run another procedure adding traffic volume. Using MuMIn functions you could evaluate the weight and importance of several different subsets of data. You can account for traffic in the same averaging model. I did not understand why present distinct models.

Validity of the findings

Introduction:
Line 43: change “have been built”
Line 54: I recommend removing “distinct”. It is not necessary for the sentence. It seems that you should compare to other taxa to say that their life traits are distinct.
Line 89: Clarify what “density” refers to. Traffic density? Road density? Polecats density?
Lines 92 – 94: here again, the sequence of objectives does not match with the abstract sequence.

Results
Line 213 – 219: Reading the results, I understood why authors use Principal Component Analysis to synthesize all land cover classes in a few components. It should be clarified in the methods. Why did they use principal components with land variables and not use the original value in the GLMM since those models can account for different data subsets?
Line 226: It is important to show the seven models. It could be in the supplementary material.

Discussion:
Line 236: There is no discussion about temporal findings. Why?
Line 237: Was there any assumption to separate models in habitat and traffic models? Why did the authors not consider all variables in the same model?
Line 274-275: When authors mentioned “fragmentation that results in habitat loss”, it seems that they are assuming “fragmentation per se” in line 271. I recommend distinguishing those terms. I indicate this review: https://doi.org/10.1146/annurev-ecolsys-110316-022612
Line 278-279: Do they avoid roads or do they avoid crossing roads?
Line 283: Do these locations refer to which locations? where polecats occur, cross, or are roadkilled?
Line 284: Reducing the speed limit is a good option for mitigation measures depending on several factors. For this species, which are speeders, I am not sure that this action works. Is there any reference to sustaining that?
Line 285: Revise the reference. Is “Studies” right?
Line 317: This is a result and should be in that section. Is this finding based on the analysis of it is just an observational result? It is crucial because all this paragraph is based on that finding.
Line 323: If unclassified roads are small rural roads, you could use this meaning in the figure to facilitate reading comprehension.

Additional comments

Abstract:
Lines 30 – 37: I recommend that methods and results have the same sequence. Example: lines from 30 to 33, firstly showed the component analysis and regression and after temporal analysis. The results (line 33) begin with temporal findings. To reader follow the ideas easier, I suggest using the same sequence.

Lines 28, 34, and 36: mortality is a population parameter. As we do not know exactly the effect of each casualty in the population, I recommend using casualties or fatalities instead of mortality. Or use WVC since you used it in the introduction. It is important to standard the terms in the whole manuscript.

---

## Round 0.2 · Minor Revisions

Apologies, but I was sick with COVID and was unable to address the reviewers' suggestions to give to you an answer.

Overall, this manuscript had improved substantially since the first round of revisions. Unfortunately, you have overlooked a few aspects of reviewers' suggestions with respect to the results. Some of the discussion remains highly speculative and not well linked to the results or the aims; again, this needs revision before publication.

Finally, reading the manuscript, please take into account the suggestions and changes made by the reviewers, which will help to organize the paper and make it easier to read.

Reviewer 1 ·

Basic reporting

The paper is reasonably well written, although there are some confused sentences, articulations and poor grammar. A little more attention to detail would go along way to improving the readability of the manuscript, with but frequent minor edits.

Experimental design

There are some aspects which require correction as they have been mis-represented / mis-described by the authors.

Line 191 -192: "analysed using a Generalized Additive Mixed Model (GAMM)"

This statement is incorrect. In the original round of reviews, the other reviewer asked what the random factor in the model was – a point I missed. The authors response was that they used the gamm function in mgcv to model correlations.

This is completely appropriate to do, however, in doing so, they are indeed not actually running a GAMM. They are simply running a GAM, with a correlation argument through a function labelled “gamm”. The authors need to rectify the text that this is not in-fact a “mixed” model with random effects. It is just a Generalised Additive Model.

Line 207: Need to explain the reasoning behind the inclusion of Moran's I as a fixed effect. This was well explained in the response to the reviewers, but not translated into the ms.

Line 208-9: Given your response to reviews. Why not simply state here “all variables had VIF <2? Stating this as something that “is done” implies that some of the above variables may not be retained.

Line 217: Again, you mean GLM here, you have not have random effect structures.

Validity of the findings

A few amendments are required:

Line 252: The authors state they have reported confidence intervals, but these are clearly not lower and upper confidence values; I suspect they are the slope estimate and the Standard error. To report Confidence intervals these need to be calculated. estimate + or - 1.96*SE.

Line 270 (but also throughout): Because the authors haven't actually reported confidence intervals they haven't used appropriate caution in their interpretation. I.e. some effects are clear (confidence intervals won't overlap zero); here however, this effects 95% CI does overlap zero, hence more caution needs to be applied with it's interpretation. Common in modern day statistics is the use of the word "trend" rather than "effect".

Line 274-5: This is a rather bizzare final line that is not associated with the model described in the paragraph (as far as I can tell). This information would be better suited to the first paragraph of the results.

Line 285 - 293: This point is rather convoluted and confused, and contradictory in places. The authors first state that increased male mortality could lead to a skewed sex ratio - yet previously they point out a polygynous mating strategy which makes this unlikely to manifest. Indeed they then highlight this themselves on line 290 where they highlight the greater importance of females. They also become overly-speculative here by extending their discussion to genetic structuring. As these information are well beyond the scope of their study I would suggest it should not be mentioned in the opening paragraph of a discussion - as per my previous revision - this content should be later in the discussion, more concise, and clear in the point being made. In my view this section could form a single sentence focussed on future research needs.

Line 357 - 361: This information should be in the results, not at the back end of the discussion.

Line 369: the authors state that the relationship between roadkill and traffic volume is well documented, yet provide no reference...

Figure 2: This plot would be more valuable to readers if on the original scale, rather than the scaled data. The overall aesthetic of the plot could also be improved.

Additional comments

A thorough revision of grammar throughout would further benefit the readability of this ms.

·

Basic reporting

The authors did a good review with many improvements for a better understanding of the manuscript, especially regarding the order of subjects and methods.

Still, there are a few suggestions. Two are related to the same comments already made in the previous review and not considered by the authors.The others are related to minor writing adjustments and citations.

Line 52-53: Those citations could be revised as they are not the best for that sentence, especially Jacobson et al. 2016.
Line 77: I did understand the explanation about why authors chose to maintain the word ‘distinct’ as they did not compare to other taxa. I read the explanation, but it is unclear for me.
Line 118: The citation “Červinka et al. 2015” can be removed as it is also cited in the end of the sentence (line 119).
Line 139: I understand authors included “by professionals” in the sentence to make clear that records were verified by specialist. However, in my opinion, this is not enough to assume that unverified records could be considered as true polecats (lines 152-153). The question related to the reliability of response data still persists.
Line 240-241-242: Mortality can be replaced by WVC
Line 250-253-256-265-269-277: in all these lines, revise the term ‘mortality’. It is needed to revise this term in the whole manuscript.
Lines 332-333: it seems that only fragmentation that causes habitat loss can create barrier to movement. I respectfully disagree with the authors’ explanation and suggest removing “that causes habitat loss”. As I mentioned before, fragmentation is not always related to habitat loss.
Line 350: Replace Jacobsen by Jacobson. The same in line 357.

Experimental design

no comment

Validity of the findings

no comment

---

## Round 0.3 · accepted · Accept

I agree with the changes and justifications presented by the authors and I am happy to tell you that the paper is ready to be accepted in PeerJ.